# Comparative In Vitro Killing by Pradofloxacin in Comparison to Ceftiofur, Enrofloxacin, Florfenicol, Marbofloxacin, Tildipirosin, Tilmicosin and Tulathromycin against Bovine Respiratory Bacterial Pathogens

**DOI:** 10.3390/microorganisms12050996

**Published:** 2024-05-15

**Authors:** Joseph M. Blondeau, Shantelle D. Fitch

**Affiliations:** 1Department of Clinical Microbiology, Royal University Hospital and Saskatchewan Health Authority, Saskatoon, SK S7N 0W8, Canada; shantelle.fitch@saskhealthauthority.ca; 2Pathology and Laboratory Medicine and Ophthalmology, Departments Biochemistry, Microbiology and Immunology, University of Saskatchewan, Saskatoon, SK S7N 0W8, Canada

**Keywords:** killing, pradofloxacin, bovine respiratory disease, drug concentrations, antimicrobials, MPC

## Abstract

Pradofloxacin is the newest of the veterinary fluoroquinolones to be approved for use in animals—initially companion animals and most recently food animals. It has a broad spectrum of in vitro activity, working actively against Gram-positive/negative, atypical and some anaerobic microorganisms. It simultaneously targets DNA gyrase (topoisomerase type II) and topoisomerase type IV, suggesting a lower propensity to select for antimicrobial resistance. The purpose of this study was to determine the rate and extent of bacterial killing by pradofloxacin against bovine strains of *Mannheimia haemolytica* and *Pasteurella multocida*, in comparison with several other agents (ceftiofur, enrofloxacin, florfenicol, marbofloxacin, tildipirosin, tilmicosin and tulathromycin) using four clinically relevant drug concentrations: minimum inhibitory and mutant prevention drug concentration, maximum serum and maximum tissue drug concentrations. At the maximum serum and tissue drug concentrations, pradofloxacin killed 99.99% of *M. haemolytica* cells following 5 min of drug exposure (versus growth to 76% kill rate for the other agents) and 94.1–98.6% of *P. multocida* following 60–120 min of drug exposure (versus growth to 98.6% kill rate for the other agents). Statistically significant differences in kill rates were seen between the various drugs tested depending on drug concentration and time of sampling after drug exposure.

## 1. Introduction

Bovine respiratory disease (BRD) is a complex pulmonary infection in cattle with pathogenesis initiated by viral infection and subsequently a secondary bacterial infection [1] of which *Mannheimia haemolytica* and *Pasteurella multocida* are primary pathogens [2]. In veterinary medicine, antimicrobial agents are used for the treatment of BRD with approved drugs representing beta-lactam, fluoroquinolone, macrolide and phenicol classes of compounds [3]. Antimicrobials may be used for metaphylaxis—the treatment of a group of animals without evidence of infection—or treatment of acute or chronic infection [4].

Antimicrobial agents have different characteristics and in vitro kill experiments have been used to compare the rate and extent of bacterial killing by various drugs and between compounds within the same drug class [5,6,7]. Differentiation of agents as bactericidal or bacteriostatic is based on a ≥3 log_10_ or ≤2 log_10_ reduction in viable cells, respectively [8], yet the clinical value of these designations has been repeatedly debated for decades and clinical evidence showing a treatment difference in outcome remains lacking [9,10]. Regardless, the use of these designations for comparing reductions in viable cells in in vitro assays still has value and allows comparison between drugs for killing bacteria—including rapid killing at clinically relevant drug concentrations—which may have clinical implications. Leekha and colleagues indicated bactericidal agents are preferred in serious infections to achieve a rapid cure [11]. Coetzee and colleagues investigated the association between antimicrobial drug classifications—bacteriostatic and bactericidal—and a number of variables including clinical outcomes [3]. Interestingly, they found that while BRD mortality was not impacted by the order in which treatment by drug classes was given i.e., bacteriostatic versus bactericidal, other non-clinical parameters were, including average daily weight gain and choice quality grade at slaughter. As such, continuing to investigate antimicrobial agents, bacterial inhibition and killing remains relevant.

Pradofloxacin is the newest fluoroquinolone approved in veterinary medicine and at present has been used for the treatment of infections in dogs and cats [12]. Pradofloxacin is a broad-spectrum antimicrobial agent with in vitro activity against Gram-positive and -negative bacteria and atypical bacteria such as *Chlamydia* species and *Mycoplasma* species organisms. Pradofloxacin also has activity that works against anaerobic bacteria [12]. It simultaneously targets DNA gyrase and topoisomerase IV in both Gram-negative and Gram-positive bacteria, and has been suggested to have a lower propensity for the selection of resistant bacterial subpopulations due to this “dual” enzyme-targeting characteristic [12,13]. In previous investigations from our laboratory, we compared in vitro killing by pradofloxacin and other antimicrobial agents against companion animal pathogens (*Escherichia coli*, *Staphylococcus pseudintermedius*, *Enterococcus faecalis* and *Proteus mirabilis*) [6,7,14] and showed rapid bactericidal activity.

In vitro susceptibility testing determines the minimum inhibitory concentration (MIC) following exposure of 10^5^ colony-forming units per milliliter (CFU/mL) of bacteria to the antimicrobial agent [15]. Mutant prevention concentration (MPC) testing determines the drug concentration required to block the growth of the least susceptible cells present in ≥10^9^ CFU bacterial cell densities exposed to the antimicrobial agent [16,17]. We were interested in comparing the rate and extent of killing by pradofloxacin as compared to ceftiofur, enrofloxacin, florfenicol, marbofloxacin, tildipirosin, tilmicosin and tulathromycin against bovine strains of *M. haemolytica* and *P. multocida* using four clinically relevant drug concentrations—MIC, MPC, maximum serum and maximum tissue drug concentrations. Pradofloxacin was rapidly bactericidal killing between 61 and 100% of bacterial cells at the MPC, C_max_ and maximum tissue drug concentrations following 30 min of drug exposure. For *M. haemolytica*, 99.99–100% of cells were killed following 5–10 min of pradofloxacin exposure at the maximum serum and tissue drug concentrations. For *P. multocida*, 82–94% of cells were killed following 60 min of exposure to pradofloxacin. Statistically significant differences were seen between the fluoroquinolones tested and the other agents depending on the drug concentration and time point.

## 2. Materials and Methods

### 2.1. Bacterial Strains

Three non-duplicate clinical isolates each of *M. haemolytica* and *P. multocida* collected from field trials in the USA were used. Organism identification was by matrix-assisted laser desorption ionization–time of flight (MALDI-TOF) (BioMerieux, St. Laurent, QC, Canada), and was confirmed by Vitek II (BioMerieux, St. Laurent, QC, Canada). Isolates were cultured on tryptic soy agar containing 5% sheep red blood cells (BA) (Oxoid, Nepean, ON, Canada) in O_2_ at 35–37 °C for 18–24 h. Single colonies were selected and transferred to skim milk and stored frozen at −70 °C. No pre-selection criteria favoured the inclusion of organisms with specific susceptibility to any drug tested; however, each isolate had to be susceptible to each agent based on currently available recommended susceptibility MIC breakpoints [18].

### 2.2. Antimicrobial Compounds

Pure substance enrofloxacin and pradofloxacin were obtained from Bayer Animal Health (Elanco as of 2020) and prepared as per the manufacturers’ instructions. Ceftiofur (Zoetis, Kirkland, QC, Canada), florfenicol (Merck, Kirkland, QC, Canada), marbofloxacin (Vetoquinol, Laval Trie, QC, Canada), tildipirosin (Merck, Kirkland, QC, Canada), tilmicosin and tulathromycin (Zoetis, Kirkland, QC, Canada) were purchased commercially and prepared in accordance with the manufacturer’s directions. Fresh stock solutions or samples stored at −70 °C were used for each experiment.

### 2.3. MIC Testing

MIC testing followed the recommended procedure by the Clinical and Laboratory Standards Institute [19]. Briefly, thawed isolates were sub-cultured twice on BA and incubated in O_2_ for 18–24 h at 35–37 °C. Mueller–Hinton Broth (MHB) (Difco Laboratories, Detroit, MI, USA) containing 2-fold drug concentration increments was added to 96-well micro dilution trays. Drug concentration ranges from 0.001 to 128 µg/mL were used, and the range varied depending on the specific drug tested. *M. haemolytica* and *P. multocida* suspensions equal to a 0.5 McFarland standard were diluted in MHB to a final inoculum of 5 × 10^5^ cfu/mL and added to microtiter trays. They were incubated for 18–24 h at 35–37 °C in O_2_, following which the lowest drug concentration preventing visible bacterial growth was recorded as the MIC. The American Type Culture Collection (ATCC) strains *Enterococcus faecalis* 29212, *Escherichia coli* 25922, *Staphylococcus aureus* 29213 and *Pseudomonas aeruginosa* 27853 were tested with each MIC assay to ensure the assays were within acceptable performance ranges.

### 2.4. MPC Testing

Using a modified MPC protocol, 5 BA plates per isolate were inoculated for confluent growth and incubated for 18–24 h at 35–37 °C in O_2_, following which the complete plate contents of bacterial growth were transferred to 100 mL of MHB and incubated for 18–24 h at 35–37 °C in O_2_ [20,21]. Following this, cultures were estimated to have concentrations of ≥3 × 10^9^ cfu/mL by spectrophotometric readings (600 nm) ≥0.3 (Thermo Scientific Genesys 10s vis, Mississauga, ON, Canada) and by colony counts. Aliquots of 100 µL containing ≥10^9^ cfu were applied to antimicrobial-containing BA plates over a range of drug concentrations from one dilution below the measured MIC to the determined MPC value. Drug plates were used within 1 week of preparation. Inoculated plates were incubated as described above with examination for growth at 24 and 48 h. The lowest drug concentration preventing growth (48 h) was the MPC. Each experiment included the 4 ATCC control strains summarized above.

### 2.5. Kill Experiments

*M. haemolytica* and *P. multocida* isolates were incubated for 18–24 h at 35–37 °C in O_2_ on BA, following which an inoculum was transferred to MHB and incubated at 35–37 °C in O_2_ for 2 h and spectrophotometric readings of ≥0.3 verified cell densities ≥10^9^ cells/mL [20]. The subsequent adjustment of inocula to achieve cell densities of 10^5^ cfu/mL was in MHB, to which an antimicrobial agent was added. Colony counts at time 0 for *M. haemolytica* and *P. multocida* were, respectively (cfu/mL), MH—ceftiofur 2.7–6.57 × 10^5^, enrofloxacin 1.87–8.4 × 10^5^, florfenicol 1.43–8.27 × 10^5^, marbofloxacin 1.8–7.3 × 10^5^, pradofloxacin 1.63–7.03 × 10^5^, tildipirosin 1.63 × 10^5^–1.1 × 10^6^, tilmicosin 2.1–8.27 × 10^5^, tulathromycin 2.03–9.17 × 10^5^; PM—ceftiofur 41.8 × 10^5^–1.3 × 10^6^, enrofloxacin 6.3 × 10^5^–1.33 × 10^6^, florfenicol 4.23 × 10^5^–1.2 × 10^6^, marbofloxacin 4.87 × 10^5^–1.4 × 10^6^, pradofloxacin 7.5 × 10^5^–3.6 × 10^6^, tildipirosin 4.37 × 10^5^–1.9 × 10^6^, tilmicosin 2.63 × 10^5^–1.1 × 10^6^ and tulathromycin 2.23 × 10^5^–1.45 × 10^6^.

Antimicrobial concentrations used in kill experiments were based on measured MIC or MPC drug concentrations for each antimicrobial agent used against each strain. The maximum serum (C_max_) and maximum tissue (T_issuemax_) drug concentrations were from published studies or reports for ceftiofur, enrofloxacin, florfenicol, marbofloxacin, pradofloxacin, tildipirosin, tilmicosin and tulathromycin (Table 1). Killing (log_10_ reduction in viable cells and percentage of the organism killed) was recorded at 5, 10, 15, 20, 25, 30, 60, 120 and 180 min following drug exposure by culturing aliquots on drug-free blood agar plates incubated for 18–24 h at 35–37 °C in O_2_. Bacterial killing was quantified by measuring the reduction in viable cell count from time 0 to the count at time 5 min after drug exposure and so on. Three separate aliquots were sampled at each time frame and the results were averaged, as were the results for the 3 strains of each genus so that each data point on the log_10_ reduction graphs represents the average of 9 individual measurements (i.e., measurements in triplicate and averaged for 3 strains).

### 2.6. Statistical Analysis

A statistical analysis of the data was performed using a repeated-measures ANCOVA for each drug data set, with fixed effects consisting of drug and drug-by-time interaction [7]. In each model, the CFU count at time 0 was included as a covariate and a compound symmetric covariance structure was used. The transformed square root CFU counts were used to achieve a normal distribution. Bonferroni adjustments for multiple comparisons were made. The least square means were back-transformed and presented as log_10_ means. Values of *p* ≤ 0.05 were considered significant for all analyses.

## 3. Results

The MIC and MPC values for each bacterial strain tested against the eight antimicrobial agents are shown in Table 1. Against the *M. haemolytica* strains tested, MIC values were lowest for ceftiofur and the three fluoroquinolones tested (0.008–0.063 µg/mL) and ranged from 0.031 to 2 µg/mL for florfenicol and from 0.5 to 4 µg/mL for tildipirosin, tilmicosin and tulathromycin. MPC values were lowest for pradofloxacin (0.031 µg/mL) and ranged from 0.063 to 0.125 µg/mL for ceftiofur, enrofloxacin and marbofloxacin and from 2 to 8 µg/mL for florfenicol, tildipirosin and tulathromycin. MPC values for tilmicosin ranged from 4 to ≥32 µg/mL.

For the *P. multocida* strains tested, MIC values were lower for ceftiofur and the fluoroquinolone compounds (0.002–0031 µg/mL) followed by florfenicol and tulathromycin (0.25–0.5 µg/mL), tildipirosin (0.5–1 µg/mL) and tilmicosin (2–4 µg/mL). The MPC values for pradofloxacin were 0.031–0.25 µg/mL as compared to ceftiofur and the other fluoroquinolones (0.031–0.25 µg/mL), florfenicol (0.5–1 µg/mL), tildipirosin (4 µg/mL), tilmicosin (4–32 µg/mL) and tulathromycin (1–2 µg/mL).

### 3.1. Mannheimia haemolytica

No significant differences were seen in the killing rate of *M. haemolytica* strains by any drugs at the MIC drug concentrations following the first 60 min of drug exposure (Figure 1). Following 120 min of drug exposure, more cells were killed by pradofloxacin (1.34 log_10_ 90.3% kill rate) than by tildipirosin (growth *p* = 0.0171). Following 180 min of drug exposure at the MIC drug concentration, more cells were killed; by ceftiofur (0.94 log_10_ 74.7% kill rate) than by tildipirosin (growth, *p* < 0.0001) or tulathromycin (growth, *p* = 0.0005); by enrofloxacin (0.55 log_10_ 49.9% kill rate) than by tildipirosin (growth *p* < 0.0001) or tulathromycin (*p* = 0.0054); by marbofloxacin (0.17 log_10_) than by tildipirosin (*p* = 0.0016); and by pradofloxacin (2.14 log_10_ 96.8% kill rate) than by florfenicol (growth, *p* = 0.0104), tildipirosin (*p* < 0.0001) and tulathromycin (*p* < 0.0001).

At the MPC drug concentrations tested, significant differences in kill rates were not seen between any drugs following the first 30 min of drug exposure (Figure 2). Following 60 min of drug exposure, more cells were killed by enrofloxacin (2.1 log_10_ 96.3% kill rate) than by ceftiofur (0.03 log_10_ 6.9% kill rate *p* = 0.0194), tildipirosin (growth *p* = 0.0033) and tulathromycin (growth *p* = 0.0350). Pradofloxacin (2.4 log_10_ 99.2% kill rate) killed more cells than tildipirosin (*p* = 0.0120). Following 120 min of drug exposure, enrofloxacin (3.3 log_10_ 99.8% kill rate) killed more cells than florfenicol (0.15 log_10_ 25.3% kill rate *p* = 0.0274), tildipirosin (growth *p* < 0.0001) and tilmicosin (0.25 log_10_ 13.7% kill rate *p* = 0.0150). Marbofloxacin (3.1 log_10_ *p* = 0.0008) and pradofloxacin (3.9 log_10_
*p* < 0.0001) killed more cells than tildipirosin. Following 180 min of drug exposure, ceftiofur (0.95 log_10_ 87.1% kill rate *p* < 0.0001), enrofloxacin (3.8 log_10_ 99.9% kill rate *p* < 0.0001), florfenicol (0.50 log_10_ 56.1% kill rate *p* = 0.0006), marbofloxacin (3.2 log_10_ 99.9% kill rate *p* < 0.0001), pradofloxacin (4.4 log_10_ 99.9% kill rate *p* < 0.0001), tilmicosin (0.69 log_10_ 36.5% kill rate *p* = 0.0204) and tulathromycin (0.42 log_10_ 54.1% kill rate *p* = 0.0032) killed more cells than tildipirosin (growth).

At the maximum serum drug (C_max_) concentrations tested, significant differences in killing rates between the compounds tested were not seen following the first 5 min of drug exposure; however, pradofloxacin (6.6 log_10_ 99.9% kill rate) showed a trend toward a significant difference as compared to tildipirosin (growth, *p* = 0.0561) (Figure 3). Pradofloxacin (5.3–5.7 log_10_ 100% kill rate) killed more cells than tilmicosin (growth), following 10 (*p* = 0.0190) and 25 min (*p* = 0.0286) of drug exposure and pradofloxacin (5.7 log_10_ 100% kill rate) killed more cells than tildipirosin (growth *p* = 0.0181) or tilmicosin (growth *p* = 0.0191), following 60 min of drug exposure. Ceftiofur (1.1 log_10_ 92–97.8% kill rate, *p* = 0.0354–0.0129), enrofloxacin (4.1 log_10_ 99.9% kill rate, *p* = 0.0004–0.0001), marbofloxacin (5.5 log_10_ 100% kill rate, *p* = 0.0005–0.0001) and pradofloxacin (5.7 log_10_ 100% kill rate, *p* < 0.0001 for both comparisons) killed more cells than tildipirosin (growth) and tilmicosin (growth), following 120 min of drug exposure. Ceftiofur (1.7 log_10_ 97.8% kill rate, *p* = 0.0402–<0.0001), enrofloxacin (4.4 log_10_ 99.9% kill rate, *p* = 0.0039–<0.0001), marbofloxacin (5.5 log_10_ 10% kill rate, *p* = 0.0045–<0.0001) and pradofloxacin (5.7 log_10_ 100% kill rate, *p* = 0.0003–<0.0001) killed more cells than tildipirosin (growth), tilmicosin (growth) and tulathromycin (growth), following 180 min of drug exposure. Florfenicol (1.1 log_10_ 88.4% kill rate, *p* < 0.0001 for both comparisons) killed more cells than tildipirosin or tilmicosin, following 180 min of drug exposure. Finally, a significant difference (*p* = 0.0092) was seen between tilmicosin and tulathromycin following 180 min of drug exposure despite organism growth in the presence of both drugs.

At the maximum tissue drug (T_issuemax_) concentrations and following 5–60 min of drug exposure, pradofloxacin 2.76–5.54 log_10_, 99.1–100% kill rate (*p* = 0.009–<0.0001) and enrofloxacin 1.22–3.90 log_10_, 75.9–99.9% kill rate (*p* = 0.0015–<0.0001) killed more cells than ceftiofur (growth—0.34 log_10_, growth to 52.14% kill rate) (Figure 4). Enrofloxacin (1.22–4.36 log_10_, 75.93–99.9% kill rate, *p* = 0.009–<0.0001) killed more cells than florfenicol (growth—0.23 log_10_ growth—40.36% kill rate), following 5–120 min of drug exposure and more cells 1.22–4.67 log_10_ 75.93–99.9% kill rate (0.0434–<0.0001), than tildipirosin (growth—0.67 log_10_, growth—71.38% kill rate) and tulathromycin (growth—0.58 log_10_, growth 46.5% kill rate) killed, following 5–180 min of drug exposure. Pradofloxacin (2.76–5.54 log_10_, 99–100% kill rate, 0.0018–<0.0001) killed more cells than tildipirosin (growth—0.20 log_10_, growth 32.55% kill rate), following 5–120 min of drug exposure, and more than tulathromycin following 5–180 min of drug exposure.

### 3.2. Pasteurella multocida

There were no significant differences in the killing rate of *P. multocida* strains by any drug at the MIC drug concentration following 120 min of drug exposure (Figure 5). Following 180 min of drug exposure, a significant difference in the bacterial killing rate was seen between ceftiofur (growth) and enrofloxacin (1.36 log_10_, 95.6% log kill rate, *p* < 0.0001) and between ceftiofur and pradofloxacin (1.38 log_10_, 95.7% kill rate, *p* < 0.0001). A significant difference in kill rate was seen between enrofloxacin and florfenicol (growth, *p* = 0.0092), and between enrofloxacin and tulathromycin (growth, *p* < 0.0001). Pradofloxacin killed more cells than florfenicol (*p* = 0.0029), which killed more than tulathromycin (*p* < 0.0001).

For the MPC drug concentration, no significant differences were seen in the killing rate of *P. multocida* by any of the drugs tested following 25 min of drug exposure (Figure 6). Following 30 min of drug exposure, more cells were killed by tulathromycin (0.65 log_10_, 72.8% kill rate) than by florfenicol (growth, *p* = 0.0017). Following 60 min of drug exposure at the MPC drug concentration, more bacterial cells were killed by pradofloxacin (0.75 log_10_, 82.3% kill rate) than by ceftiofur (0.02 log_10_, 4.2% kill rate, *p* = 0.441), tilmicosin (0.97 log_10_, 81.61% kill rate, *p* = 0.0242), tulathromycin (1.25 log_10_, 67.53% kill rate, *p* < 0.0001) and florfenicol (0.03 log_10_, 5.8% kill rate, *p* = 0.0033). A significant difference was seen between the killing rates of florfenicol and tildipirosin (0.68 log_10_, 77.6% kill rate, *p* = 0.0083), tilmicosin 0.97% log_10_, 81.6% kill rate, *p* = 0.0021) and tulathromycin (*p* < 0.0001). Following 120 min of drug exposure, significant differences were seen in bacterial killing rates between ceftiofur (0.021 log_10_, 4.6% kill rate), enrofloxacin (0.90 log_10_, 87.0% kill rate, *p* = 0.0134), marbofloxacin (0.91 log_10_, 86.8% kill rate, *p* = 0.0023), pradofloxacin (1.34 log_10_, 95.2% kill rate, *p* < 0.0001), tildipirosin (1.16 log_10_, 92.5% kill rate, *p* < 0.0001), tilmicosin (1.37 log_10_, 94.9% kill rate, *p* < 0.0001) and tulathromycin (2.0 log_10_, 98.5% kill rate, *p* < 0.0001). A significant difference in killing was also seen between florfenicol (0.13 log_10_, 24.2% kill rate) and marbofloxacin, pradofloxacin, tildipirosin, tilmicosin and tulathromycin (*p* values from 0.0003 to <0.0001 for all comparisons). Following 180 min of drug exposure, more cells were killed by enrofloxacin (1.22 log_10_, 94.0% kill rate, *p* = 0.0009), marbofloxacin (1.3 log_10_, 94.4% kill rate, *p* < 0.0001), pradofloxacin 1.71 log_10_, 98.1% kill rate, *p* < 0.0001), tildipirosin (1.54 log_10_, 96.7% kill rate, *p* < 0.0001), tilmicosin (2.32 log_10_, 99.1% kill rate, *p* < 0.0001), tulathromycin (2.73 log_10_, 99.4% kill rate, *p* < 0.0001) than ceftiofur (0.05 log_10_, 9.7% kill rate). A statistically significant difference was seen between the killing rate of florfenicol (0.28 log_10_, 41.3% kill rate) versus marbofloxacin (*p* = 0.0082), pradofloxacin (*p* = 0.0009), tildipirosin (*p* = 0.0001), tilmicosin (*p* = 0.0005) and tulathromycin (*p* < 0.0001).

For the C_max_ drug concentration, there were no differences in killing rate by any agent following 5 min of drug exposure (Figure 7). Following 10 min of drug exposure, statistically significant differences in killing rate were seen between ceftiofur (0.02 log_10_, 4.7% kill rate), enrofloxacin (0.19 log_10_, 35.8% kill rate, *p* = 0.0155), marbofloxacin (0.34 log_10_, 51.4% kill rate, *p* < 0.0001) and pradofloxacin (0.45 log_10_, 64.0% kill rate, *p* < 0.0001). Following 60 min of drug exposure, more cells were killed by pradofloxacin (1.38 log_10_, 94.1% kill rate) than by tilmicosin (growth, *p* = 0.0280). Following 120 min of drug exposure, more cells were killed by ceftiofur (1.04 log_10_, 89.9% kill rate), enrofloxacin (0.96 log_10_, 88.0% kill rate, *p* < 0.0001), florfenicol (1.23 log_10_, 92.7% kill rate, *p* < 0.0001), marbofloxacin (0.94 log_10_, 88.0% kill rate, *p* < 0.0001) and pradofloxacin (1.85 log_10_, 98.6% kill rate, *p* < 0.001) than by tilmicosin (growth). More cells were killed by tildipirosin (0.27 log_10_, 43.7% kill rate, *p* = 0.0519) and tulathromycin (0.68 log_10_, 78.3% kill rate, *p* < 0.0001), than by tilmicosin. Following 180 min of drug exposure, more cells were killed by ceftiofur (1.0 log_10_, 89.3% kill rate, *p* < 0.0001) than by tilmicosin (growth). More cells were killed by enrofloxacin (1.2 log_10_, 93.2% kill rate, *p* < 0.0001), florfenicol (1.5 log_10_, 96.5% kill rate, *p* < 0.001), marbofloxacin (1.4 log_10_, 95.8% kill rate, *p* < 0.001), pradofloxacin (2.2 log_10_, 99.3% kill rate, *p* < 0.0001) and tulathromycin (1.4 log_10_, 95% kill rate, *p* < 0.0001) than by tilmicosin (growth).

At the maximum tissue drug concentration, more cells were killed by pradofloxacin (0.3–0.4 log_10_, 49.6–57.1% kill rate, *p* = 0.0158) than by tildipirosin (growth, 0.1 log_10_, 1.44% kill rate, *p* = 0.0103), following 10 and 15 min of drug exposure, respectively (Figure 8). Following 20 min of drug exposure, more cells were killed by pradofloxacin (0.52 log_10_, 68.1% kill rate) than by ceftiofur (growth, *p* = 0.0405) and tildipirosin (0.04 log_10_, 7.1% kill rate, *p* = 0.0006). Following 25 min of drug exposure, more cells were killed by pradofloxacin (0.6 log_10_, 72.4% kill rate) and tulathromycin (0.54 log_10_, 72.6% kill rate) than by florfenicol (0.1 log_10_, 17.4% kill rate, *p* = 0.0011–0.0138). More cells were killed by tulathromycin than by tildipirosin (0.1 log_10_, 20.1% kill rate, *p* = 0.0257), and more cells were killed by pradofloxacin than by tildipirosin (0.1 log_10_, 20.2% kill rate, *p* = 0.0019). Following 30 min of drug exposure, more cells were killed by pradofloxacin (1.2 log_10_, 73.1% kill rate) and tulathromycin (0.78 log_10_, 58.2% kill rate) than by ceftiofur (0.04 log_10_, 33.7% kill rate, *p* = 0.0370 and *p* = 0.010, respectively). More cells were killed by pradofloxacin and tulathromycin (0.8 log_10_, 58.2% kill rate) than by florfenicol (0.1 log_10_, 12.7% kill rate, *p* = 0.0010, *p* < 0.0001). Tulathromycin killed more cells than tildipirosin (0.2 log_10_, 30.4% kill rate, *p* = 0.0010). Following 60 min of drug exposure, pradofloxacin (1.2 log_10_, 92.9% kill rate) and tulathromycin (1.3 log_10_, 95.0% kill rate) killed more cells than ceftiofur (0.2 log_10_, 33.7% kill rate, *p* < 0.0001 for both comparisons). Pradofloxacin killed more cells than enrofloxacin (0.3 log_10_, 52.2% kill rate, *p* = 0.0004), and tulathromycin (1.3 log_10_, 95.0% kill rate) killed more cells than enrofloxacin (*p* < 0.0001). Pradofloxacin (*p* = 0.0041) and tulathromycin (*p* = 0.0003) killed more cells than florfenicol (0.4 log_10_, 49.8% kill rate). Tulathromycin killed more cells than tildipirosin (0.5 log_10_, 66.2% kill rate, *p* = 0.0146). Following 120 min of drug exposure, a significant difference was seen between enrofloxacin (0.7 log_10_, 77.2% kill rate) and tulathromycin (1.9 log_10_, 98.6% kill rate, *p* = 0.0147).

## 4. Discussion

Dagan and colleagues argued that the eradication of bacteria associated with respiratory tract infections was an important aim of antimicrobial therapy, and was necessary for clinical cure and the prevention of the spread of antimicrobial resistant clones [22]. In this report, killing rates of *M. haemolytica* and *P. multocida* isolates by eight different antimicrobial agents showed variability in killing rates, with four clinically relevant drug concentrations. Leekha et al. [11], in reviewing the general principles of antimicrobial agents, indicated time-dependent antibiotics have relatively slow bactericidal action, whereas concentration-dependent agents have enhanced bactericidal activity. As such, we might expect a difference in performance in agents over the duration of the drug exposure in kill assays.

Pradofloxacin is a dual-targeting, advanced generation fluoroquinolone approved for use in veterinary medicine [23] and most recently in food animals (April, 2024). It has a broad spectrum of in vitro activity against Gram-negative, Gram-positive, atypical and anaerobic bacteria [24]. It is bactericidal and results in rapid reductions in bacterial cells at clinically important drug concentrations [6,7,14]. The simultaneous targeting of two enzymes critical for DNA replication (DNA gyrase (topoisomerase II) and topoisomerase IV) has been argued to reduce the likelihood for resistance selection [12]. While other veterinary fluoroquinolones (i.e., enrofloxacin and marbofloxacin) target both the aforementioned enzymes, the primary target in Gram-negative bacteria is DNA gyrase, and in Gram-positive bacteria it is topoisomerase IV, whereas pradofloxacin is equal in its targeting of both of these enzymes. In this study, we compared killing rates by pradofloxacin and seven other antimicrobial agents (ceftiofur, enrofloxacin, florfenicol, marbofloxacin, tildipirosin, tilmicosin and tulathromycin) against two primary bacterial pathogens—*M. haemolytica* and *P. multocida*—associated with bovine respiratory disease. Drug concentrations tested included those that are clinically relevant: MIC, MPC, maximum serum and maximum tissue drug concentrations. Pradofloxacin was rapidly bactericidal against the bacterial strains tested, and the extent of the bacterial killing rate was faster as the drug concentration increased. At the MIC drug concentration, pradofloxacin achieved a 1.3–2.14 log_10_ reduction in viable cells (90.3–96.8% kill rate), following 120–180 min of drug exposure as compared to 2.4–39 log_10_ reduction (99.2–99.9% kill rate) following 60–120 min of drug exposure at the MPC drug concentration against *M. haemolytica* strains. At the C_max_ and maximum tissue drug concentration, a 2.8—4.63 log_10_ (99.9% kill rate) reduction in viable cells was seen following 5 min of drug exposure. For *P. multocida*, exposure to pradofloxacin resulted in a 0.6–1.4 log_10_ reduction (68.9–95.7% kill rate) in viable cells following exposure to the MIC drug concentration for 120–180 min. At the MPC drug concentration, a 0.8–1.7 log_10_ reduction (82–98% kill rate) was seen following 60–180 min of drug exposure. At the C_max_ drug concentration, a 0.9–1.1 log_10_ reduction (87.7–91.6% kill rate) was seen following 25–30 min of drug exposure, which increased to a 1.3–2.2 log_10_ (94.1–99.5% kill rate) reduction following 60–180 min of drug exposure. At the maximum tissue drug concentration, 73% of cells were killed following 30 min of drug exposure, which increased to a 97.7–99.1% kill rate following 120–180 min of drug exposure. The log_10_ reduction for the other fluoroquinolones tested were less than those seen for pradofloxacin but not statistically different. Pradofloxacin (and the other quinolones tested) resulted in statistically more killing than the non-fluoroquinolone agents tested and this was not unexpected as the other agents were either time dependent compounds and/or bacteriostatic. We have previously reported on the in vitro killing rate of *M. haemolytica* strains by enrofloxacin, florfenicol, tilmicosin and tulathromycin and over a range of bacterial densities [25]. The results from this investigation are consistent with those previously reported. Ceftiofur, as expected, showed time-dependent killing rates with the highest percentages of killing (89.3–97.8% kill rate at the maximum serum and tissue drug concentrations) following 180 min of drug exposure. Tildipirosin had lower percentages for the killing rate of *M. haemolytica* at 71.4% at the maximum tissue concentration following 180 min of drug exposure. For *P. multocida*, tildipirosin killed 70.1–96.7% of strains following 180 min of drug exposure at the MPC, maximum serum and maximum tissue drug concentration.

In vitro measurements, as they are reported here, cannot be extrapolated to clinical outcomes; however, such measurements provide a comparison between drugs, under controlled conditions, on killing rates of bacteria. While it has been argued that the differentiation between bacteriostatic and bactericidal drugs is not clinically relevant, the definitions do help with comparing drugs based on the log_10_ reduction (and percent kill rate) in viable cells over time. Boswell et al. defined a bactericidal agent as resulting in a ≥3 log_10_ reduction in viable cells, whereas a bacteriostatic agent resulted in a ≤2 log_10_ reduction in viable cells [26]. Rapid reductions in viable cells may predict shorter courses of therapy; however, this would need confirmation in clinical trials investigating the length of therapy.

We have previously reported the bactericidal activity of pradofloxacin (and comparator agents) against companion animal pathogens [6,7], using clinically relevant drug concentrations such as those utilized in this study. For example, with *S. pseudintermedius*, pradofloxacin killed 90–99% of cells following 20–180 min of drug exposure at the maximum serum drug concentration. At the maximum tissue drug concentration, pradofloxacin killed 66–87% of viable cells following 5–10 min of drug exposure and 97–>99% of cells were killed following 60–120 min, respectively, of drug exposure. For *E. coli* strains exposed to the maximum serum concentration of pradofloxacin, 48–>99% of cells were killed following 5–15 min, respectively, of drug exposure. Similarly, following exposure of *E. coli* strains to the maximum tissue concentration of pradofloxacin, 83–>99% of cells were killed following 5–15 min, respectively, of drug exposure. The results from this study showing rapid killing rates of pradofloxacin are consistent with the data summarized above. We have also previously reported on the in vitro killing rate of *M. haemolytica* by enrofloxacin, tilmicosin and tulathromycin [25], but with higher bacterial densities than those reported here. Regardless, killing by the aforementioned agents was consistent between studies.

Coetzee and colleagues [27] reported on the likelihood for the selection of drug-resistant bacteria based on the order in which bacteriostatic or bactericidal antibiotics were given. For example, for treatment of bovine respiratory disease, protocols using a bacteriostatic drug (tulathromycin) for first treatment followed by a bactericidal drug (ceftiofur) for second treatment were associated with a higher frequency of a resistant BRD pathogen isolation. Interestingly, use of ceftiofur first followed by tulathromycin resulted in recovery of fewer *M. haemolytica* resistant isolates.

In a follow-up publication, Coetzee et al. [3] investigated the association between bacteriostatic versus bactericidal drug selection for the treatment and retreatment of BRD and non-clinical parameters—particularly health performance and carcass quality. Mortality differences were not seen based on drug class selection. The probability of BRD cases requiring four or more treatments compared to three treatments was greater in calves receiving bacteriostatic/bactericidal or bacteriostatic/bacteriostatic drug regimes for first/second treatments compared to those receiving bactericidal/bactericidal for first/second treatments (*p* < 0.001). Calves receiving bactericidal/bactericidal for first/second treatments had increased average daily weight gain compared to those receiving bacteriostatic/bactericidal or bacteriostatic/bacteriostatic treatments (*p* < 0.001). Calves receiving bactericidal/bactericidal treatments had a higher probability of a choice quality grade at slaughter compared to bacteriostatic/bactericidal-treated animals (*p* = 0.037). Zhang et al. [28] commented that for concentration-dependent agents, the kill rate gradually increases as drug concentrations increase, whereas for time-dependent drugs the kill rate is smaller and increases with time. Our results in this report are consistent with the results from Zhang et al. These same authors indicated that limitations with MIC testing and the rise in antimicrobial resistance dictate that the other measurements—such as MPC—require ongoing investigation.

## 5. Conclusions

In summary, pradofloxacin is the newest of the veterinary fluoroquinolones approved for use in both companion and food animals. Against bovine strains of *M. haemolytica* and *P. multocida*, pradofloxacin was bactericidal and, depending on drug concentration and time, pradofloxacin resulted in a rapid reduction in bacterial numbers. Pradofloxacin appears to be a valuable addition to antimicrobial agents for the treatment of BRD.

## Figures and Tables

**Figure 1 microorganisms-12-00996-f001:**
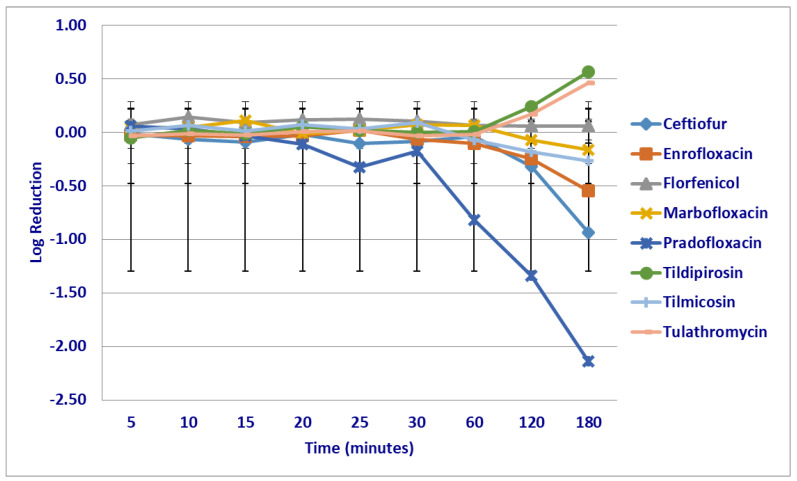
Log reduction in *Mannheimia haemolytica* (Isolates Averaged) by eight antimicrobial agents at the minimum inhibitory concentration.

**Figure 2 microorganisms-12-00996-f002:**
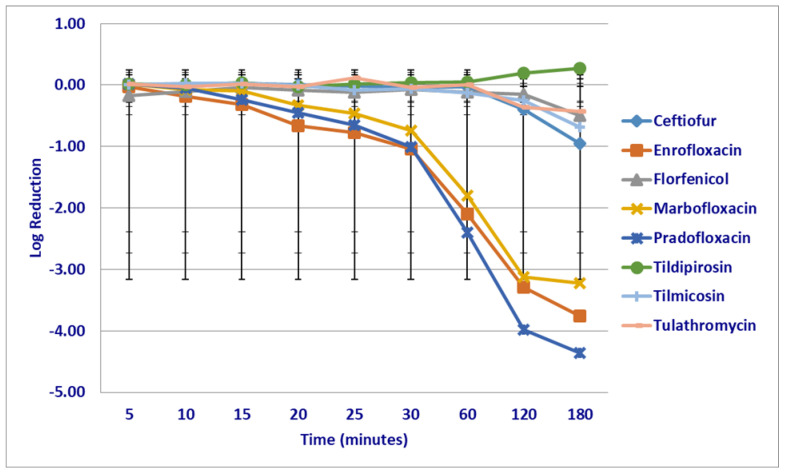
Log reduction in *Mannheimia haemolytica* (Isolates Averaged) by eight antimicrobial agents at the mutant prevention concentration.

**Figure 3 microorganisms-12-00996-f003:**
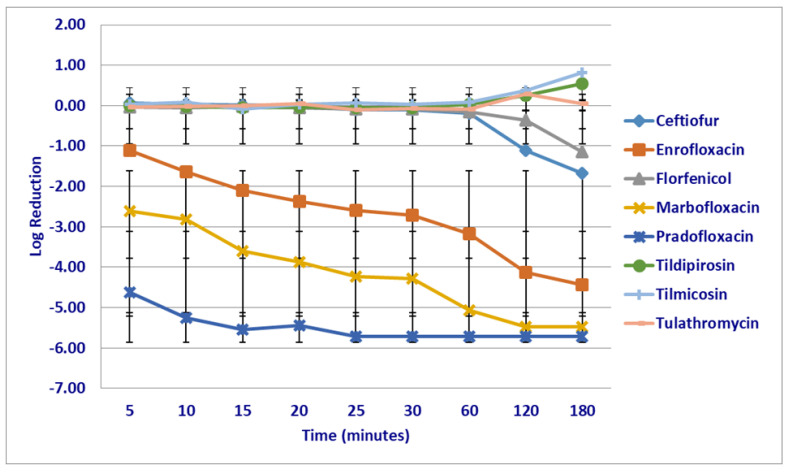
Log reduction in *Mannheimia haemolytica* (Isolates Averaged) by eight antimicrobial agents at the maximum serum drug concentration.

**Figure 4 microorganisms-12-00996-f004:**
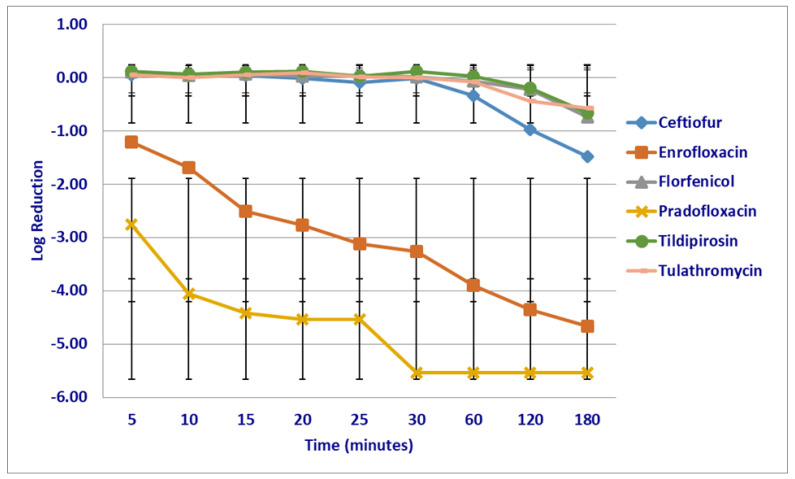
Log reduction in *Mannheimia haemolytica* (Isolates Averaged) by six antimicrobial agents at the maximum tissue drug concentration.

**Figure 5 microorganisms-12-00996-f005:**
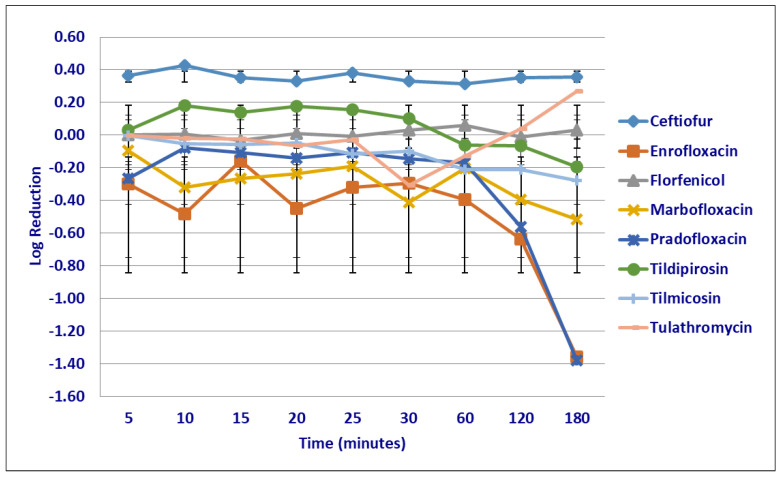
Log reduction in *Pasteurella multocida* (Isolates Averaged) by eight antimicrobial agents at the minimum inhibitory concentration.

**Figure 6 microorganisms-12-00996-f006:**
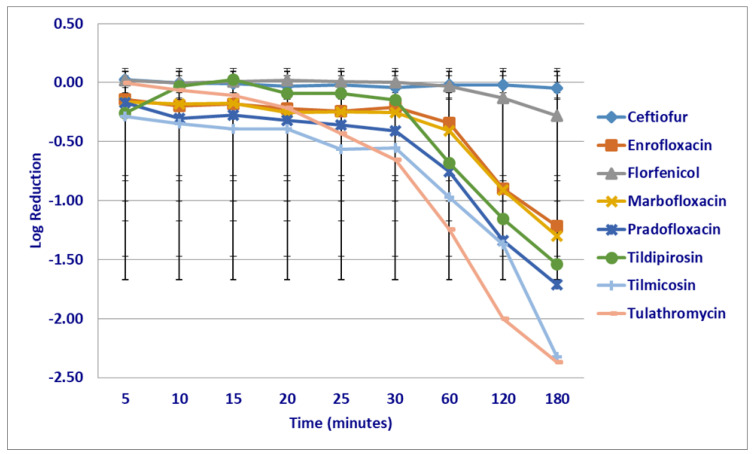
Log reduction in *Pasteurella multocida* (Isolates Averaged) by eight antimicrobial agents at the mutant prevention concentration.

**Figure 7 microorganisms-12-00996-f007:**
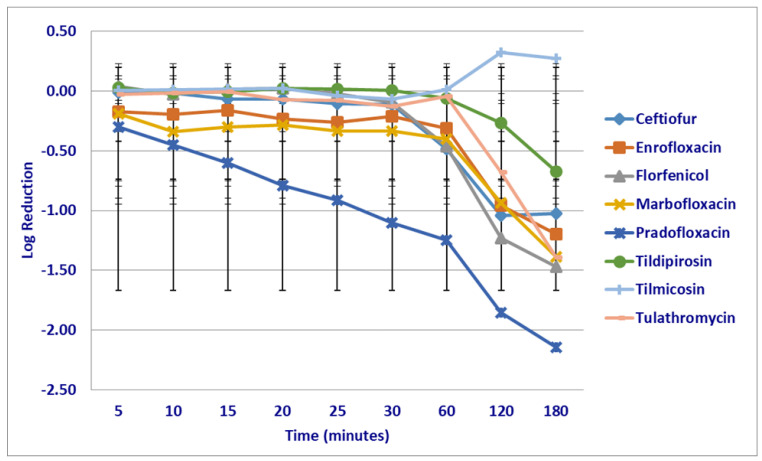
Log reduction in *Pasteurella multocida* (Isolates Averaged) by eight antimicrobial agents at the maximum serum drug concentration.

**Figure 8 microorganisms-12-00996-f008:**
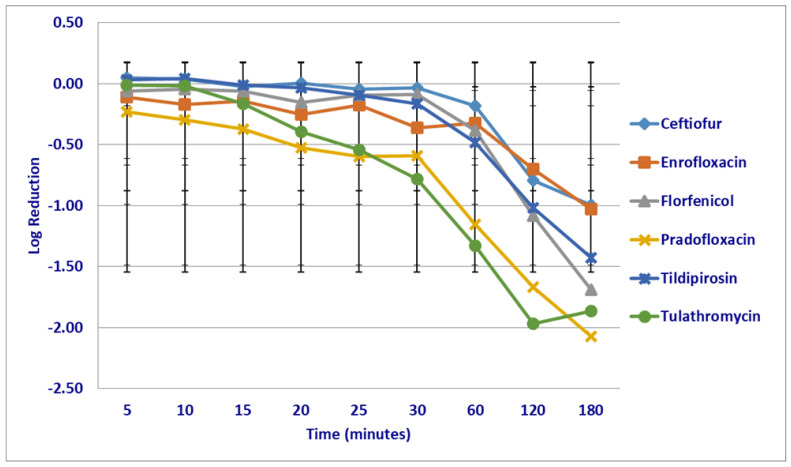
Log reduction in *Pasteurella multocida* (Isolates Averaged) by six antimicrobial agents at the maximum tissue serum drug concentration.

**Table 1 microorganisms-12-00996-t001:** Comparative MIC, MPC and therapeutic drug concentration values for 8 antimicrobial agents.

	Isolates						C_max_ (µg/mL)	T_issuemax_ (µg/mL)
	MIC	MPC	MIC	MPC	MIC	MPC		
*M. haemolytica*	#36 (170)		#17 (9–83)		#13 (54–78)			
Ceftiofur	0.008	0.125	0.008	0.125	0.031	0.063	6.9	2.64
Enrofloxacin	0.063	0.5	0.016	0.125	0.016	0.125	1.9	4.6
Florfenicol	0.031	2	2	2	2	4	4.5	2.94
Marbofloxacin	0.016	0.063	0.016	0.063	0.106	0.063	1.5	NT
Pradofloxacin	0.016	0.031	0.016	0.031	0.008	0.031	2.64	0.81
Tildipirosin	0.5	2	1	2	1	2	0.767	14.77
Tilmicosin	0.5	4	0.5	16	4	≥32	0.25	NT
Tulathromycin	0.5	2	0.5	2	1	8	0.6	3.2
*P. multocida*	#5		#6		#14			
Ceftiofur	0.002	0.125	0.002	0.125	0.002	0.25	6.9	2.64
Enrofloxacin	0.008	0.063	0.004	0.063	0.008	0.031	1.9	4.6
Florfenicol	0.5	1	0.25	1	0.5	0.5	4.5	2.94
Marbofloxacin	0.016	0.125	0.008	0.125	0.016	0.25	1.5	NT
Pradofloxacin	≤0.008	0.031	0.031	0.031	0.004	0.25	2.64	0.81
Tildipirosin	1	4	0.5	4	0.5	4	0.767	14.77
Tilmicosin	4	32	2	8	2	4	0.25	NT
Tulathromycin	0.5	2	0.25	1	0.5	1	0.6	3.2

MIC = minimum inhibitory concentration; MPC = mutant prevention concentration; C_max_ = maximum serum drug concentration, T_issuemax_ = maximum tissue drug concentration; µg/mL = microgram per milliliter; and NT = not tested.

## Data Availability

The data that support the findings of this study are available from the corresponding author upon reasonable request.

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
