# Peer review of "Comparative In Vitro Killing by Pradofloxacin in Comparison to Ceftiofur, Enrofloxacin, Florfenicol, Marbofloxacin, Tildipirosin, Tilmicosin and Tulathromycin against Bovine Respiratory Bacterial Pathogens"

_microorganisms, 2024, doi:10.3390/microorganisms12050996_

Round 1

Reviewer 1 Report

Comments and Suggestions for Authors

Prafofloxacin, the newest veterinary fluoroquinolone, has demonstrated its usefulness in companion and food animals. However, the article in question lacks meticulous logic, rigorous writing, and requires careful revision.

In the Introduction, specifically in lines 68-93, the fourth part should clearly state the experiment's purpose. Additionally, the summary of the experimental results should be moved to the discussion section.

The MIC experimental method lacks a description of different drug concentrations and how the MIC is determined. This information needs to be included.

The overall experimental method description is overly complex and should be simplified for better reader comprehension.

In line 185, there is an incorrect statement regarding the MPC range of tilmicosin. This should be revised with careful attention.

The figures lack uniformity in their formatting. It is recommended to standardize the format, ensuring consistency in the presence or absence of borders.

Table 1 does not adhere to strict formatting guidelines. The numbers of the two types of strains, as well as the positions of MIC and MPC, should be made uniform.

Following statistical analysis of the data, the significance of the differences should be clearly indicated in the figures.

The experimental section of the study compares prafloxacin with the concentrations of four clinically relevant drugs. However, the discussion section solely focuses on the results of prafloxacin, neglecting the other experimental findings. It is important to address all relevant results in the discussion.

Comments on the Quality of English Language

English can be read but it is not enough smooth and comfortable

Reviewer 2 Report

Comments and Suggestions for Authors

The main objective was to determine the rate and extent of bactericidal activity of pradoxacin against Mannheimia haemolytica bovis strains and Pasteurella multocida, compared with several other drugs, using four clinically relevant drug concentrations: minimal inhibitory and mutagenic prophylactic drug concentrations, maximum serum and maximum tissue drug concentrations. This is an interesting manuscript, but it needs improved article quality and professional English editing to be published.

1. The abstract contains only vague references to results. There are no quantitative values or even strong relative statements. Revise the abstract to include more description in the abstract and more detail about the changes. Abstract rewrites the description of the results and the last sentence .

2. There is a lot of irregularity and unprofessionalism in the writing of the article. Some descriptions are hard to understand.

3. please revise the keywords and choose the ones that are more representative of this experiment.

4. At the end of the introduction, please add purpose and significance.

5. Lack of detailed description of material methods and key experimental treatments. For example, the process of strain identification.

6. materials and methods section lacks details of methods.

7. statistics: it is not clear how many experiments were conducted.

8. there are some grammatical errors and spelling mistakes, please check the whole manuscript and revise.

9. please use a three-line table to represent the results.

Comments on the Quality of English Language

This is an interesting manuscript, but it needs improved article quality and professional English editing to be published.

Reviewer 3 Report

Comments and Suggestions for Authors

The article is very interesting as it deals with the issue of new treatments for bacteria, in this case for use in veterinary medicine.

The summary is well-explained and gives us an idea of what the article is about.

The introduction is too long. I believe the last paragraph should be moved to the discussion section (lin.

 I also feel that this section lacks the inclusion of a description of the working hypothesis and a list of objectives/objective.

The materials and methods are very well described and could be replicated by any other research group, adding value to the study.

The results are clear, and the tables and graphs are appropriate.

Regarding the discussion section, while it's well-structured by first stating the most significant results and then comparing them with other studies, I find this part to be too extensive. Once a study is cited, it shouldn't include such exhaustive data from other studies because it makes the reading too broad.

The conclusions contain too many numbers and percentages, which makes them repetitive with the results. I would recommend summarizing the key findings without relying heavily on numerical details.

The bibliography section is well-organized and contains consistently appropriate articles of interest for the study.
